# Treatment outcomes among children and adolescents with extensively drug–resistant (XDR) and pre–XDR tuberculosis: Systematic review and meta–analysis

**Jayadeep Patra**[1,2]*, **Hyacinth Irving**[1], **Pranshu Maini**[3], **Jady Liang**[4], **Anwesh Patra**[2], **Mandar Paradkar**[5], **Jurgen Rehm**[1,2]

**1** Dalla Lana School of Public Health, University of Toronto, Toronto, Ontario, Canada, **2** Centre for Addiction and Mental Health, Toronto, Ontario, Canada, **3** Faculty of Health Sciences, McMaster University, Hamilton, Ontario, Canada, **4** Department of Physiology, University of Toronto, Toronto, Ontario, Canada, **5** Johns Hopkins Center for Infectious Diseases in India, Pune, Maharashtra, India

* Jaydeep.Patra@gmail.com

## Abstract

Extensively drug–resistant (XDR) and pre-XDR- tuberculosis (TB) account for approximately a third of pediatric MDR–TB cases globally. Clinical management is challenging; recommendations are based on limited evidence. We assessed the clinical outcomes for children and adolescents treated for XDR–and pre–XDR–TB. We performed a systematic review and meta–analysis of published studies reporting treatment outcomes for children and adolescents with XDR–and pre–XDR–TB. MEDLINE, EMBASE, Scopus, Web of Science, Google Scholar, and trial registries up to 31 December 2023 were searched. Eligible studies included children and adolescents aged <18 years with XDR–or pre–XDR–TB. The primary outcome was treatment success, defined as a composite of cure and treatment completion. Secondary outcomes were death, failure/ lost to follow–up and adverse events. We identified 34 population-based studies and 14 case studies, which reported treatment outcomes for a total of 656 patients. Treatment durations ranged from 6 to 27 months; follow–up after treatment completion ranged from 2 months to 4 years. The pooled estimate for treatment success in XDR–and pre-XDR-TB was 88·9% (95%CI: 59·7–100%) and 65·4% (95%CI: 27·7–100%), respectively. Drug adverse effects were reported in 56.4% (95%CI: 9.9–100%) and 68.2% (95%CI: 0–100%) of children respectively. Few childhood XDR- and pre-XDR-TB cases are reported. The relatively good treatment outcomes in children compared to adults may be partly due to publishing bias. Drug adverse effects are common.

## Introduction

The global burden of TB among children and adolescents remains a critical public health concern. In 2022, an estimated 1.3 million TB cases occurred among children aged 0–14 years,

**Data Availability Statement:** All relevant data are within the paper and its Supporting Information files.

**Funding:** The authors received no specific funding for this work.

**Competing interests:** The authors have declared that no competing interests exist.

accounting for 12% of all TB cases worldwide [1]. A modeling exercise has suggested that globally 25,000 to 32,000 children developed MDR-TB in 2021. Of these MDR-TB cases, 29% were pre-XDR and 5% were XDR [2]. MDR-TB is defined as disease caused by M. tuberculosis resistant to both rifampicin and isoniazid, whereas extensively drug–resistant (XDR) TB is defined as MDR–TB with additional resistance to any fluoroquinolone and at least one second–line injectable drug) [1]. In cases where the MDR isolates are resistant to either a fluoroquinolone or an injectable, but not both, the resistance pattern is termed as pre–XDR [3].

Extrapulmonary tuberculosis (EPTB) poses a particular challenge in pediatric populations. In 2019, EPTB accounted for about 16% of 7.1 million reported TB cases globally [4]. Children are more susceptible to EPTB compared to adults, with factors such as young age and developing immunity contributing to this increased risk [5]. The manifestations of EPTB in children can be diverse and often non-specific, making diagnosis challenging. Common forms include lymph node TB, TB meningitis, and disseminated TB, each presenting unique diagnostic and treatment challenges. However, the specific global burden of EPTB among children remains undetermined, highlighting a significant gap in our understanding of pediatric TB epidemiology.

The WHO recommends second-line drug susceptibility testing (DST) for all patients with MDR–TB in order to diagnose further resistance and prevent delay with appropriate treatment [1,2]. Both MDR- and XDR–TB can be acquired, usually because of poor treatment management, or transmitted, implying primary infection with a resistant strain. It is therefore important to do first–line DST in both new and previously treated patients. It is estimated that globally only 12% of new TB cases and 58% of previously treated cases receive first–line DST (to at least isoniazid and rifampicin) to identify MDR–TB [1].

The classification of XDR- or pre-XDR TB via second-line DST is completed in only 24% of all confirmed MDR–TB, thus raising a concern that diagnoses and treatment for such cases remain low [1]. Reports of XDR–and pre–XDR–TB in children and adolescents are even more limited. The paucibacillary nature of pediatric TB disease, the barriers inherent in obtaining suitable specimens, and the low sensitivity of microbiological diagnostics and DST in children continues to further exacerbate the challenges involved in identifying drug–resistant TB, thus delaying treatment and adversely affecting prognosis [1]. Drug–resistant disease in children is transmitted primarily from infectious adults with drug-resistant TB, hence affirming the importance of contact tracing [2].

Recent years have seen major and unprecedented setbacks in TB diagnosis and therapeutic developments. The COVID-19 pandemic led to reversals in the number of newly diagnosed TB cases in 2020 and 2021, severely impacting global TB treatment targets [1].

Despite a recovery in 2022, none of the treatment targets set for 2018–2022 at the first UN high- level meeting on TB were achieved. The cumulative number of people treated for TB between 2018 and 2022 reached 34 million, including 2.5 million children, falling short of the 5-year targets of 40 million and 3.5 million, respectively [1]. The shortfall is even more pronounced in the treatment of MDR/RR-TB in children, where only 21,600 children (19% of the target) received treatment against a WHO target of 115,000 for the period 2018–2022 [1].

Even for adults, treatment guidelines for the management of drug–resistant TB are based on limited evidence and many of the studies on which the recommendations were based excluded patients with XDR–TB. In children, there is even less evidence to support recommendations for drug-resistant TB treatment [1]. With few drugs available and low treatment coverage, XDR–TB represents a significant global health concern, and thus, exploring the treatment options for this disease is an important step towards serving the global efforts to eliminate TB [1].

Despite the unforeseen setbacks, a few recent advancements in TB treatment offer renewed hope that stands to improve patient outcomes [6]. These include the introduction of shorter (4-month) regimens for drug-susceptible TB, which can improve treatment adherence and reduce the burden on patients and health systems [6]. All-oral regimens for MDR/RR-TB have been developed, effectively avoiding the use of injectable agents associated with significant adverse events, particularly in children [6]. The expanded use of novel and highly efficacious drugs like bedaquiline and delamanid in younger children has also broadened treatment options for pediatric drug-resistant TB [6]. Additionally, the development of child-friendly formulations of TB drugs and therapies has greatly facilitated effective administration of treatment to pediatric populations [6].

However, significant challenges remain in diagnosing and treating these conditions in young children. These include the difficulty in obtaining appropriate samples for diagnosis, the lower sensitivity of diagnostic tests in children, the lack of pediatric formulations for some second-line drugs, and the complexities of managing drug-resistant TB in the context of HIV co- infection and malnutrition, which are common comorbidities in high-burden settings.

With few drugs available and low treatment coverage, XDR–TB represents a significant global health concern. Exploring the treatment options for this disease is an important step in serving the global efforts to eliminate TB. The WHO End TB Strategy aims to end the global TB epidemic by 2035, but achieving this goal will require intensified research and innovation to provide new tools and interventions, particularly for the management of drug-resistant TB in all populations, including children and adolescents.

Given these complex challenges and the urgent need to improve outcomes for children with drug-resistant TB, there is a critical need for a comprehensive review of treatment strategies and outcomes for XDR- and pre-XDR-TB in pediatric populations. We conducted a systematic review and meta–analysis to study the treatment and subsequent outcomes of XDR– and pre–XDR–TB in children and adolescents, aiming to address this crucial gap in knowledge and contribute to the global efforts to combat TB in one of the most vulnerable affected populations.

## Methods

### Search strategy

This meta–analysis was done according to PRISMA guidelines (see S1 Text) [7]. We conducted a systematic literature search for publications using the following electronic databases up to 31st February 2024: MEDLINE, EMBASE, Scopus, Web of Science, Google Scholar, and trial registries with combinations of the search terms "tuberculosis", "multidrug resistance", "multidrug–resistant", "extensively drug–resistant", "XDR*", "Pre–XDR", "treatment outcomes", and "children", both as exploded MESH headings and free–text terms. Additionally, the online archives of the *International Journal of Tuberculosis and Lung Disease* and conference abstracts from the *International Union Against Tuberculosis and Lung Disease* (2004–2024) were reviewed [8]. We applied no language restriction to our search of abstracts.

An overview of the study protocol is provided in study protocol (S2 Text).

### Study selection

References from the selected studies were also hand–searched to ensure that relevant studies were not omitted. The contents of the abstracts or full–text manuscripts identified during the literature search were reviewed independently by 2 investigators (PM and AP) to determine whether they met the criteria for inclusion. When there were discrepancies between investigators for inclusion or exclusion, a third reviewer (JP) was consulted. We excluded studies that

reported on adults only, discussed diagnostic techniques, treatment guidelines, the prevalence and risk factors, or outcomes for MDR–or DR–TB only. Studies were eligible if they included children and/or adolescents (<18 years) within a defined treatment cohort including cases of XDR and/or pre–XDR–TB. In anticipation of a scarcity of data, case series of 1 or more patients were also included. Following WHO clinical guidelines, cases were included if they were diagnosed either with microbiologically confirmed XDR- or pre-XDR-TB or had clinical TB disease and had been in close contact with a source case who had confirmed XDR- or pre-XDR- TB.

### Data extraction, quality assessment, and study characteristics

All treatment outcomes were defined in accordance with WHO classification systems. Our primary outcome was treatment success, which included cure, treatment completion, or both. Secondary outcomes were the treatment categories of death, failure, as well as adverse events. 'Failure' included patients who had treatment failure or were lost to follow–up or transferred out. Patients who remained on treatment or had unknown treatment outcomes by the end of the study were excluded from our analysis. Authors of relevant studies were contacted for clarification and additional information when necessary. To assess the methodological quality of the included studies, we used a modified Newcastle–Ottawa scale [9].

To examine differences in treatment outcome, we collected data on patient and study characteristics (setting, age, gender, method of diagnosis, HIV status and other comorbid conditions, previous treatment, contact history, and the study outcome definitions), the treatment regimen (drugs used and DST results, duration of treatment), and length of follow–up after treatment completion or discontinuation. In order to assess the study quality, we also extracted information on variables that could have affected treatment success: microbiological confirmation, DST–based individualized treatment, use of injectable drugs and/or fluoroquinolones, admission to hospital at the initiation of treatment, and direct observation throughout the treatment. All study and patient-related characteristics were extracted using the data extraction tool developed in excel (refer to S1 Appendix).

### Quantitative data synthesis

Pooled proportions and 95% confidence intervals (CI) for each TB outcome were reported in a random effects meta–analysis. Treatment success was estimated conservatively, with all children lost to follow–up regarded as treatment failures. We stabilized the variance of the raw proportions using a Freeman–Tukey–type arcsine square–root transformation [10] and calculated pooled estimates using (a) DerSimonian and Laird random–effects model and (b) using a Bayesian random–effects model with Monte Carlo Markov chain simulations with non–informative priors and Gibbs sampling for the mean and variance parameters with 95% CI on all outcomes; [11,12] convergence was visually assessed using Brooks–Gelman criteria [13,14].

We calculated the $\tau^2$ statistic to assess the proportion of overall variation attributable to between study heterogeneity [15]. We used meta–regression to evaluate whether effect size estimates were significantly different by study characteristics: DST confirmatory testing, culture positivity, use of injectable drugs and use of fluoroquinolone (each ≤50% or >50% of the cohort); treatment duration (≤ 18 months or >18 months), median age (≤ 5 years or >5 years), HIV status (positive or negative in the cohort), and contact with a MDR/XDR TB patient (yes or no). All statistical procedures were carried out in Stata (version 18).

## Results

### Search results

Out of 483 studies identified and screened, we obtained the full-text articles for 48. Of these, 25 studies were included in the quantitative analysis; 34 were population–based (mostly retrospective cohorts) and 14 were case studies/series (Fig 1). Although we did not restrict our search by language, all qualifying studies were reported in English. We found 18 articles in other languages (Spanish, Chinese, and Portuguese), which were excluded as they did not provide treatment outcome data on children. These 48 studies described 10702 patients, of which 81 children were included in our analysis, with the rest excluded for not meeting inclusion criteria or for being duplicated cases.

### Patient characteristics

277 children had XDR–TB and 379 had pre–XDR–TB. The cases were from a wide range of countries, with the majority (60 of 81, 74·1%) from high TB burden areas [1]. Of the 36 cases with XDR–TB and 24 cases of pre–XDR–TB who had HIV–testing performed, 115 XDR–TB (35%) and 199 pre–XDR–TB cases (54%) were HIV–positive (Table 1). Girls represented a larger proportion in both XDR–TB (28/48, 58%) and pre–XDR–TB groups (18/33, 54%).

Twenty- seven of 81 (33%) were 0–4 years of age. The median age was 8 years with a range of 1 to 14·6 years. Of the 81 children, 57 (71%) had pulmonary TB (PTB) (S1 Table), 43 (53·1%) had previously been treated for TB, and 41/81 (51%) had a known source case with MDR–or XDR–TB patient (S1 Table). Seventy-eight of 81 (96%) cases were culture and DST–confirmed; the remaining three patients were diagnosed based on their history of contact with an XDR–TB patient (Table 1).

### Treatment

In children for whom it was specified whether injectable agents or fluoroquinolones were used, 40 of 45 (89%) were prescribed at least one injectable medication and 45 of 55 (82%) were prescribed a fluoroquinolone (Table 2). Pre–XDR–and XDR–TB patients were prescribed a median of five and six drugs, respectively, and their isolates also demonstrated resistance to a median of five and six drugs, respectively (Table 2).

### Outcome

Time to culture–conversion ranged from 1 to 15 months, with a median time of 3 months (Table 2). For patients with a specified duration of treatment, the total duration was 12 months or more for 60 of 69 (57·3%) patients, with follow–up ranging from 2 months to 4 years after completion of therapy. The proportion of children with successful treatment outcomes was high in the XDR–TB patient group with a lower proportion in the pre–XDR–TB groups (88·9% 95%CI: 59·7–100 [$I^2 = 0\%$, $\tau^2 = 0.0$]; 65·4% 95%CI: 27·6–100 [$I^2 = 0\%$; $\tau^2 = 0.1$], respectively) with very low heterogeneity and between-study variance (Fig 2A). Rates of death and treatment failure or lost to follow-up were low; though more children with pre-XDR-TB died (19·3% 95%CI 0–57·5 vs. 10·2%, 95%CI: 0–40·4), or had greater rates of failure or default (16·7%, 95%CI 0–50·9 vs. 11·8%, 95%CI 0–40·2), as compared to children with XDR–TB.

In our subgroup analysis for all patients, we found a trend towards higher success rates for patients who were HIV negative (86·6%, 95%CI: 67·8–99), and did not have a known history of contact with a patient with MDR- or XDR-TB (89·9%; 95%CI: 71·3–100) as compared to the reference groups. Since all studies used injectables, and most studies used fluoroquinolones, DST–guided treatment regimens culture–based diagnoses, we did not have a

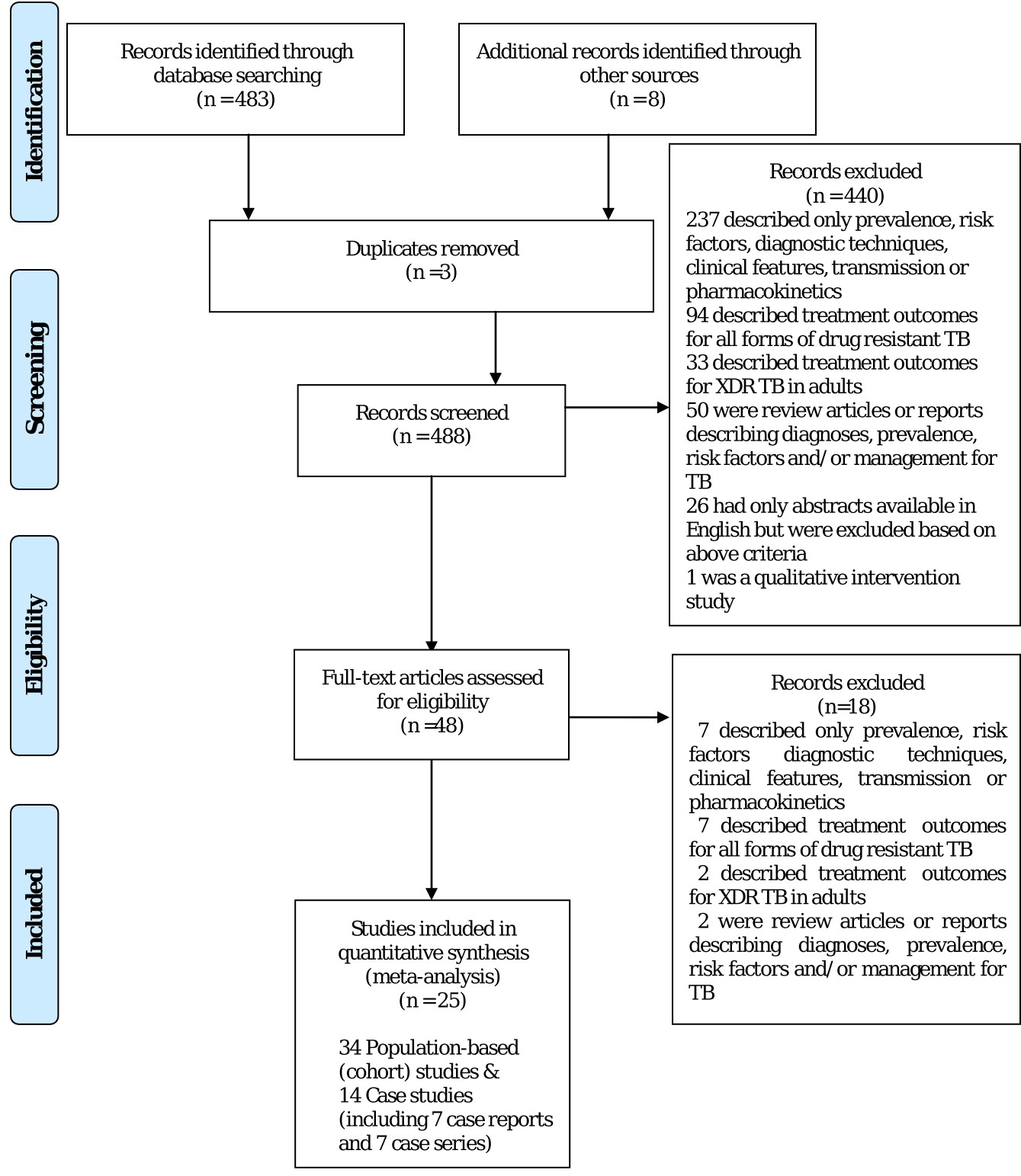

**Fig 1. Study identification and inclusion.**

Table 1. Description of included studies and demographic characteristics of patients.

| Study | Location | National TB prevalence[1] (per 10⁵) | Study period | (a) Study design (b) Study setting | Median age (years, range) | Gender (%) | Sample size | Number of cases (XDR/pre-/partial-XDR) | (a) HIV prevalence (%) (b) Other co-morbid conditions (%) | (a) Method of diagnoses‡ (b) Clinical/Radiologic presentation | Culture confirmed TB* | DST confirmation on case's sample* |
|---|---|---|---|---|---|---|---|---|---|---|---|---|
| Population-based | | | | | | | | | | | | |
| Hicks et al (2014) [16] | South Africa | 696 | 2009–2010 | (a) R Cohort (b) Facility | 8 (4–12) | M (42·9%) F (57·1%) | 84 | 16 (6 XDR, 10 pre-XDR) | (a) NS (b) NS | (a) Culture test (using 2 sputum samples), CXR, DST (b) NS | 100% | 100% |
| Kuksa et al (2014) [17] | Latvia | 57 | 2000–2010 | (a) R Cohort (b) National (Population-based, surveillance-based) | 42 (0–55+) XDR: 39 (0–55+) XDR pediatric: NS (<18) | M (76%) F (24%) XDR: M (65%), F (35%) | 30 | 7 (XDR) | (a) NS (b) NS | (a) Culture test, DST (b) NS | 100% | 100% |
| Mignone et al (2014) [18] | Ukraine/ Peru | 114/158 | 2006–2010 | (a) R Cohort (b) Facility | 5·2 (0.5–17·9) XDR: 16 (15·8–16·3) | M (50%) F (50%) XDR: F (100%) | 22 | 2 (XDR) | (a) 0% (b) None | (a) Smear and Culture test, CXR, DST (b) Fever, vomiting, chest pain | 100% | 100% |
| Seddon et al (2014) [19] | South Africa | 696 | 2009–2010 | (a) P Cohort (b) Sub-national (Community-based) | 3 (1–12) | M (46·3%) F (53·7%) | 149 | 6 (XDR) | (a) NS (b) NS | (a) Culture test, CXR, DST (b) Perihilar infiltrates, Hilar lymphadenopathy, Lobar/segmental collapse or opacification, Pleural effusion, Cavities | 50% | 50% |
| Isaakidis et al (2013) [20] | India | 195 | 2007–2013 | (a) R Cohort (b) Facility | 14·6 (10–18) | M (46%) F (54%) | 10 | 5 (1 XDR, 4 pre-XDR) | (a) 100% (b) None | (a) Smear and Culture test, DST (b) NS | 100% | 100% |
| Gegia et al (2013) [21] | United States | 4 | 2009–2011 | (a) R Cohort (b) Sub-national (Community-based) | 8 (<1–16) | M (64·4%) F (35·6%) | 45 | 3 (XDR) | (a) 0‡ (b) NS | (a) Smear and Culture test, Clinical examination, DST (b) PTB: unilateral lesions on left/right upper lobe (100%). EPTB: Peripheral lymphadenopathy (74.2%) | 100% | 100% |
| Williams et al (2013) [22] | UK/ India | 15/195 | 2006–2010 | (a) R Cohort (b) Sub-national (Community-based) | 13 (1–15) XDR: 8·4 (2–13) | M (23·5%) F (75·5%) XDR: M (67%), F (33%) | 17 | 3 (pre-XDR) | (a) 0‡ (b) None | (a) Culture/Smear test, CXR, TST, DST (b) Cough, fever, weight loss, night sweats, lethargy | 100% | 100% |

*(Continued)*

Table 1. (Continued)

| Study | Location | National TB prevalence[1] (per 10⁵) | Study period | (a) Study design (b) Study setting | Median age (years, range) | Gender (%) | Sample size | Number of cases (XDR-/pre-/partial-XDR) | (a) HIV prevalence (%) (b) Other co-morbid conditions (%) | (a) Method of diagnoses§ (b) Clinical/Radiologic presentation | Culture confirmed TB* | DST confirmation on case's sample* |
|---|---|---|---|---|---|---|---|---|---|---|---|---|
| | | | | **Among Total sample*** | | | | | | | | |
| **Rose et al (2012) [23]** | South Africa | 696 | 2007–2012 | (a) R Cohort (b) Facility | 6 (1–13) | F (100%) | 2 | 2 (1 XDR, 1 pre-XDR) | (a) 100% (b) Unilateral chronic suppurative otitis media (50%), Bronchiectasis, hearing loss, perianal warts (50%) | (a) Culture test, CXR, DST (b) Pneumonia, bronchial compression, perihilar/paratracheal lymphadenopathy, cavitation, extensive bronchiectasis | 100% | 100% |
| **Seddon et al (2012) [24]** | South Africa | 696 | 2003–2008 | (a) R Cohort (b) Facility | 4 (<2–9) | M (41·4%) F (58·6%) | 111 | 12 (5 XDR, 7 pre-XDR) | (a) NS (b) NS | (a) Culture test, CXR, Ultrasound, DST (b) Hilar lymphadenopathy or airway compression, lobar/segmental collapse or opacification, cavitation | 100% | 100% |
| **Liu et al (2011) [25]** | China | 89 | 1996–2009 | (a) R Cohort (b) Facility | 41 (2–99) XDR: NS (2–99) XDR pediatric: NS (2–14) | M (70·1%) F (29.9%) XDR: M (75%), F (25%) | 18 | 2 (XDR) | (a) 0 (b) NS | (a) Smear and Culture test, DST (b) NS | 100% | 100% |
| **Thomas et al (2010) [26]** | South Africa | 696 | 2006–2007 | (a) R Cohort (b) Facility | 7 (6–8) | M 2 (50%) F 2 (50%) | 4 | 4 (XDR) | (a) 100% (b) Kwashiorker (50%) Diarrhea (25%) | (a) Culture test, CXR, DST (b) Cavitary disease and extensive infiltrates, Perihilar lymphadenopathy | 100% | 100% |
| **Case-studies** | | | | | | | | | | | | |
| **Salazar-Austin et al 2015 [27]** | United States | 3.8 | .. | (a) Case study (b) Facility | 2 | M | 1 | 1 (XDR) | (a) 0 (b) None | (a) Culture test, Clinical signs, CT scan, DST (b) Fever, tachycardia, tachnpnoea, low body weight, left lower lobe infiltrate and hilar adenopathy | 100% | 100% |
| **Alsleben et al (2014) [28]** | South Africa | 696 | 2011–2012 | (a) Case study (b) Facility | 4 | F | 1 | 1 (XDR) | (a) 0 (b) None | (a) Clinical signs, CT scan, DST (b) Fever, abdominal discomfort, worsening headache, periorbital edema,vomiting, ataxia, meningeal enhancement | 100% | 100% |

(Continued)

Table 1. (Continued)

| Study | Location | National TB prevalence[1] (per 10⁵) | Study period | (a) Study design (b) Study setting | Median age (years, range) | Gender (%) | Sample size | Number of cases (XDR/pre-/partial-XDR) | (a) HIV prevalence (%) (b) Other co-morbid conditions (%) | (a) Method of diagnoses‡ (b) Clinical/Radiologic presentation | Culture confirmed TB* | DST confirmation on case's sample* |
|---|---|---|---|---|---|---|---|---|---|---|---|---|
| | | | | **Among Total sample*** | | | | | | | | |
| **Mohan et al (2014)** [29] | India | 195 | .. | (a) Case study (b) Facility | 3 | F | 1 | 1 (XDR) | (a) 0 (b) None | (a) Clinical signs, CT scan, DST (b) Spinal abscess, cord compression, paraparesis, kyphotic deformity in the upper back | 100% | 100% |
| **Rodrigues et al (2014)** [30] | Portugal | 29 | .. | (a) Case study (b) Facility | 10 | F | 2 | 1 (XDR) | (a) 0 (b) Pneumonia | (a) AFB sputum test, CXR, TST, DST (b) Vespertine fever, chest pain, nonproductive cough, anorexia, consolidation in the upper left lobe, endobronchic dissemination | 100% | 100% |
| **Uppuluri et al (2014)** [31] | India | 195 | .. | (a) Case study (b) Facility | 8 | F | 1 | 1 (pre-XDR) | (a) 100% (b) None | (a) AFB sputum smear test, CXR, DST (b) Cough, fever, weight loss, left lower lobe consolidation | 100% | 100% |
| **Katragkou et al (2013)** [32] | Greece | 6 | .. | (a) Case study (b) Facility | 2.5 | F | 2 | 1 (XDR) | (a) 0 (b) None | (a) Clinical signs, TST, DST (b) Fever, cought, weight loss | 100% | 100% |
| | | | | | 1.5 | M | 2 | 1 (pre-XDR) | (a) 0 (b) Epilepsy | (a) Culture test (using PCR), MRI, DST (b) Fever, cough, vomit, and tonic–clonic seizures | 100% | 100% |
| **Payen et al (2012)** [33] | Russia | 109 | .. | (a) Case study (b) Facility | 14 | F | 1 | 1 (XDR) | (a) 0 (b) None | (a) Culture test, DST (b) NS | 100% | 100% |
| **Dauby et al (2011)** [34] | Nigeria | 330 | .. | (a) Case study (b) Facility | 14 | F | 1 | 1 (XDR) | (a) 0 (b) None | (a) Culture test, CXR, DST (b) Fever, cough, anorexia, failure to thrive, lesion, severely damaged left lung | 100% | 100% |

(Continued)

 

**Table 1.** (Continued)

| Study | Location | National TB prevalence[1] (per 10^5) | Study period | (a) Study design (b) Study setting | Median age (years, range) | Gender (%) | Sample size | Number of cases (XDR/pre-/partial-XDR) | (a) HIV prevalence (%) (b) Other co-morbid conditions (%) | (a) Method of diagnoses§ (b) Clinical/Radiologic presentation | Culture confirmed TB* | DST confirmation on case's sample* |
|---|---|---|---|---|---|---|---|---|---|---|---|---|
| | | | | | | **Among Total sample*** | | | | | | |
| **Kjollerstrom et al (2011) [35]** | Portugal | 29 | .. | (a) Case study (b) Facility | 14 | M | 4 | 1 (pre-XDR) | (a) 0 (b) None | Culture test, CXR, DST | 100% | 100% |
| | | | | | 12 | F | 4 | 1 (XDR) | (a) 0 (b) Sickle cell disease | (a) Culture test, CXR, clinical signs, DST (b) Bilateral involvement, cavitation, endobronchial dissemination | 100% | 100% |
| **Shah et al (2011) [36]** | India | 195 | 2007 (Case 1) | (a) Case study (b) Facility | 2.5 | F | 3 | 3 (pre-XDR) | (a) NS (b) None | (a) Culture test, CXR, X ray of the spine, TST, DST (b) Cough and fever, left foot swelling and discharge (non-foul smelling), weight loss/ loss of appetite | 100% | 100% |
| | | | | | 6 | F | | | (a) NS (b) Malnourishment | (a) Culture test, CXR, DST (b) Cough, fever, consolidation in the left upper lobe, bilateral inguinal adenopathy, hepatomegaly | | |
| | | | | | 9.5 | M | | | (a) NS (b) None | (a) Culture test, CXR, TST, DST, Ultrasound (b) Dry cough, fever, abdominal pain, reduced appetite, non significant cervical/ inguinal/axillary lymph nodes, hepatomegaly | | |
| **Anger et al (2010) [37]** | United States | 4 | 2000–2006 | (a) Case study (b) Sub-national (Community-based) | 10 | F | 1 | 1 (XDR) | (a) 100% (b) Hepatitis B | (a) Culture test, CXR, DST (b) NS | 100% | 100% |
| **Kulkarni et al 2009 [38]** | India | 195 | .. | (a) Case study (b) Facility | 7 | M | 1 | 1 (XDR) | (a) 0 (b) None | (a) Culture test, CXR, TST, DST with Antibiogram (b) Fever, headache, lethargy, loss of appetite, upper abdominal pain | 100% | 100% |

(Continued)

 

Table 1. (Continued)

| Study | Location | National TB prevalence[†] (per 10^5) | Study period | (a) Study design (b) Study setting | Median age (years, range) | Gender (%) | Sample size | Number of cases (XDR/pre-/partial-XDR) | (a) HIV prevalence (%) (b) Other co-morbid conditions (%) | (a) Method of diagnoses¶ (b) Clinical/Radiologic presentation | Culture confirmed TB* | DST confirmation on case's sample* |
|---|---|---|---|---|---|---|---|---|---|---|---|---|
| | | | | | | | | | **Among Total sample*** | | | |
| **Schaaf et al. 2009** [39] | South Africa | 696 | .. | (a) Case study (b) Facility | <1 | F | 1 | 1 (XDR) | (a) 0 (b) None | (a) Culture test, CXR, TST, DST (b) Cough, fever, weight loss and failure to thrive, tachypenia, hepatomegaly and hilar lymphadenopathy | 100% | 100% |
| **Schluger et al 1996** [40] | United States/ Egypt | 4/26 | .. | (a) Case study (b) Facility | <2 | M | 2 | 1 (pre-XDR) | (a) 0 (b) None | (a) Culture test (gastric aspirate), TST, CXR, CT scan (b) Fever, failure ro thrive, lymphadenopathy | 100% | 100% |
| | | | | | 1 | F | | | (a) 0 (b) None | (a) Culture test, TST, CT scan, DST (b) Cough, fever, low body weight/height (10th percentile), infiltrates/calcification in the right lobe | 100% | 100% |

Note

*Represents the overall sample, including MDR/DRTB cases; information for XDR cases is shown if specified in the study

†The majority of XDR cases were culture–positive and were treated based on their DST results except for the cases in Seddon et al (2014) [19], where 50% of the XDR cases were culture–confirmed, and thus, DST was completed on 50% of cases' samples; the remaining 50% cases were treated based on the DST results from the source

‡HIV status was not known (Williams et al (2013) [22]) or tested (Gegia et al (2013) [21]) for all XDR cases. XDR: Extensively drug resistant, P cohort: prospective cohort, R cohort: retrospective cohort, NS: Not spefied, DST: Drug sensitivity test, TST: Tuberculosis Skin Test, CXR: Chest X–Ray, CT scan: Computerized Tomography.

**Table 2. Description of treatment regimen.**

| Study | Length of treatment (months)* | Treatment description | Injectables used | Fluroquinolones used | Resistance pattern | Definition of success |
|---|---|---|---|---|---|---|
| **Population-based** | | | | | | |
| **Hicks et al (2014) [16]** | 24 mos of XDR or pre-XDR TB treatment, including at least 18 mos after culture conversion | Individualized based on DST results. Drugs used: One injectable (Amk, Km or Cm) R, E, others not specified. | NS | NS | H, R, at least 1 injectable (Am/Km/Cm) and 1 FQ | Cure (WHO) |
| **Kuksa et al (2014) [17]** | 18 to 36 mos of XDR TB treatment, including 18 mos after culture conversion | Individualized based on DST. Drugs used: In addition to the MDR regimen, Mfx, Amx/Clv, Lzd, Clr and/or Imp/Cln. Administred daily in hospital, and 5–6 times/week in community or other medical facilties. | Cm | Mfx or Lfx | H, R, at least 1 injectable (Am/Km/Cm) and 1 FQ | Cure or tx completion. Cure: tx completed with negative cultures 18 months after culture conversion. Tx completed: completion of therapy with no cultures performed |
| **Mignone et al (2014) [18]** | At least 18 mos of XDR TB treatment after culture conversion | Individualized based on DST results. Drugs used: Mfx, Lzd, Amx/Clv, Cfz, R, E, Z | None | Mfx | H, R, S, Km, Z, Cs, PAS, Eto | Cure (WHO) |
| **Seddon et al (2014) [19]** | Up to 21 mos, including at least 18 mos after culture conversion | Individualized based on DST results. Drugs used: Not specified for XDR cases | NS | NS | H, R, E, Eto, Amk, Ofx | Cure or probably cured (WHO) |
| **Isaakidis et al (2013) [20]** | 3 to 18 mos of XDR or pre-XDR TB treatment, continued therapy after culture conversion | Individualized based on DST. Drugs used: Cm,Lfx,Eto,Cs, Amcl, Mfx, PAS, Z, Km, high-dose H, Cfz | Cm or Km | Lfx or Mfx | H, R, Z, E, S,, Km, PAS, Eto, Mfx, Ofx | Cure or tx completion (WHO) |
| **Gegia et al (2013) [21]** | 18.2 to 28.3 mos of XDR TB treatment, including at least 18 mos after culture conversion | Individualized based on DST results. Drugs used: Z, Cm, Lfx, prothionamide, Cs, PAS. Education, nutritional and transportation support. | Cm | Lfx | H, R, at least 1 injectable (Am/Km/Cm) and 1 FQ | Cure or tx completion (WHO) |
| **Williams et al (2013) [22]** | 18 to 30 mos | Individualized based on DST results. Drugs used: Amk, Pto, Mfx, Cs, PAS, Z, H, R, E, S | Amk | Mfx | H, R, Z, E, S, Eto, Pto, Mfx, Ofx, Amk, Km, Cm | Tx completion and clinical and radiological resolution of symptoms by the end of therapy and upto 2 years of follow-up. |
| **Rose et al (2012) [23]** | 18 (for pre-XDR) to 29 mos (for XDR), including at least 18 mos of therapy after culture conversion. Both patients were culture negative at the start of treatment for pre-XDR or XDR. | Individualized based on DST results. Drugs used: H, Z, Eto, Cpm, Trd, Clm, Amx/Clv, Lzd, PAS, Clr, Ofx, Mfx, Cfz | Cm | Mfx or Ofx | H, R, E, Amk, Km, S, Eth, Ofx, | Cure (WHO) |
| **Seddon et al (2012) [24]** | Around 24 mos of XDR or pre-XDR TB treatment, including at least 12 mos after culture conversion | Individualized for all XDR cases based on DST results. Drugs used: H, Amk or Cm, Ofx, Eto, PAS, Trd, Amx/Clv, Clr, Lzd | NS | Ofx | H, R, Amk, Ofx | Cure: negative results of 3 consecutive respiratory cultures obtained within at least 30 days of each other with persistent negative cultures, and tx completion. |
| **Liu et al (2011) [25]** | 22.3 mos of XDR TB treatment, including at least 12 mos after culture conversion | Individualized based on DST results. Drugs used: H, R, E, Z, Ofx, Lfx, S, Km, PAS, Z, Amk, Cpm, Mfx, Th, Rpt, Rfb | One of Amk/Km/Cm | One of Mfx/Ofx/Lfx | H, R, Z, E, Ofx, Lfx, S, Km, PAS, Z, Amk | Cure and Tx completion (WHO) |

*(Continued)*

**Table 2.** (Continued)

| Study | Length of treatment (months)* | Treatment description | Injectables used | Fluroquinolones used | Resistance pattern | Definition of success |
|---|---|---|---|---|---|---|
| **Population-based** | | | | | | |
| **Thomas et al (2010) [26]** | At least 24 mos of XDR TB treatment, included at least 21 mos after culture conversion | Individualized based on DST results. Drugs used: Eto, PAS, Z, Cs, Cm, E, Clr, Amx/Clv | Cm | None | H, R, S, E, Cfx, Km | Cure (WHO) |
| **Case-studies** | | | | | | |
| **Salazar-Austin et al 2015 [27]** | 18 mos of XDR TB treatment, culture negative at treatment initiation for XDR TB, but treatment continued due to poor clinical and radiologic presentation | Individualized based on DST results. Drugs used: S (4 µg/mL), Lzd, PAS, Cs, Cfz. Supplemented with vitamin B6 | None | None | H, R, Z, S (1 µg/mL), Ofx, Mfx | Culture conversion at 13 weeks, persistent negative cultures until tx completion and at follow-up, clinical and radiological improvement (weight gain, normal hearing, resolution of lung infiltrate 6 months into tx) |
| **Alsleben et al (2014) [28]** | 18 mos of XDR TB treatment, including at least 9 mos after culture conversion | Individualized based on DST results. Drugs used: Cfz, Clr, Cm, Eto, H, Lfx, Lzd, PAS, Trd | Cm | Lfx | R, H, Amk, Ofx, S, E | Culture conversion at 9 months, persistent negative cultures until tx completion and at follow-up, no signs of recurrence |
| **Mohan et al (2014) [29]** | At least 8 mos of XDR TB treatment | Individualized based on DST results. Drugs used: PAS, Eto, Cfz, Lzd, Clr, Amx/Clv. Surgery for decompression, instrumentation and deformity correction. | None | None | H, R, Z, E, S | Tx completion, neurological improvement (reduction in spinal abscess) and ability to walk without support |
| **Rodrigues et al (2014) [30]** | 18 mos of XDR TB treatment, including 15 mos after culture conversion | Individualized based on DST results. Drugs used: Lfx, PAS, Cs, Amk, Lzd, Z | Amk | Lfx | H, R, E, Z | Culture conversion at 12 weeks, persistent negative cultures until tx completion and 18 months of follow-up, clinical improvement |
| **Uppuluri et al (2014)† [31]** | At least 12 mos of pre-XDR TB treatment, culture negative at 12 mos, remains on treatment | Individualized based on DST results. Drugs used: Amk, PAS, Cfz, Lzd | Amk | None | H, R, Z, E, S, Eto, Ofx, Mfx | Culture conversion at 12 months, clinical improvement (weight gain) |
| **Katragkou et al (2013) [32]** | At least 12 mos of XDR TB treatment, including 18 mos after culture conversion. Cultures were negative prior to starting XDR TB regimen | Individualized based on DST results. Drugs used: Lzd, Cs, PAS, Rfb | None | None | H, R, Z, E, S, Amk, Lfx | Culture conversion at 5 months, persistent negative cultures, clinical and radiologic improvement |
| | 6 mos of pre-XDR TB treatment, including 5 mos after culture conversion. Child died by the 6th month due to acute heaptic insufficiency. | Individualized based on DST results. Drugs used: H, Lfx, Cm, Cs, Lzd. External CSF drainage | Cm | Lfx | H, R, Z, E | Culture conversion at 1 month, persistent negative cultures until tx completion and follow-up, normal CXR, no signs of disseminated TB |
| **Payen et al (2012) [33]** | At least 18 mos of XDR TB treatment after culture conversion | Individualized based on DST. Drugs used: Z, Cm, Cs, Lzd, Clr, M-C | Cm | None | Mfx, Amk, Pto, Cs, Rfb | Cure (WHO) |
| **Dauby et al (2011) [34]** | Around 20 mos of XDR TB treatment, including 18 mos after culture conversion | Individualized based on DST results. Drugs used: Cm, Lzd, Clr, Mem, Amx/Clv, Z, Cs. Supplemented with vitamin B6 | Cm | None | H, R, Rfb, Ofx, Eto, Z, Amk, Cs, and Pto | Culture conversion at 11 weeks, persisting negative cultures until tx completion, weight gain, reduced lesions on lungs |
| **Kjollerstrom et al (2011) [35]** | 24 mos of pre-XDR TB treatment, including 21 mos after culture conversion | Individualized based on DST results. Drugs used: Lzd, Rfb, Z, Cs, PAS, Km, Lfx | Km | Lfx | H, R, Z, Eto, S, Rfb, Eto, Cm, Amk | Culture negative at 12 weeks, persistent negative cultures at tx completion and 4 years of follow-up, absence of clinical signs |
| | 24 mos of XDR TB treatment, including 22 mos after culture conversion | Individualized based on DST results. Drugs used: Lzd, Rfb, Z, PAS, E, Cfz, Amk, Lfx | Amk | Lfx | H, R, Z, S, Rfb, Eto, Cs, PAS, Km, Ofx | Culture negative at 6 months, persistent negative cultures and no signs of peripheral neuropathy at 3 yrs of follow-up |

*(Continued)*

**Table 2.** (Continued)

| Study | Length of treatment (months)* | Treatment description | Injectables used | Fluroquinolones used | Resistance pattern | Definition of success |
|---|---|---|---|---|---|---|
| **Population-based** | | | | | | |
| **Shah et al (2011)† [36]** | NA; loss to follow up | NA; loss to follow up | NA; loss to follow-up | NA; loss to follow up | NA; loss to follow up | NA; loss to follow up |
| | At least 15 mos of pre-XDR TB treatment, culture negative at 15 mos, and remains on treatment | Individualized based on DST results. Drugs used: Z, Amk, Mfx, PAS, Eto | Amk | Mfx | H, R, E, S, Ofx | Culture negative at 15 months, clinical and radiological improvement (asymptomatic, weight gain, reduced lung consolidation) |
| | At least 15 mos of pre-XDR TB treatment, asymptomatic, and remains on treatment | Individualized based on DST results. Drugs used: PAS, Amk, Eto, Gfx. Abdominal lymphnode biopsy. | Amk | Gfx | H, R, Z, E, S, Ofx, Mfx | Clinical and radiological improvement (asymptomatic, regression of lymphnodes) |
| **Anger et al (2010) [37]** | 27 mos of XDR TB treatment, including 25 mos after culture conversion | Individualized based on DST results. Drugs used: Lzd, Z, Cm, Eto, Lfx, Ipm | Cm | Lfx | Z, Cm, Km, Amk, Eto, Cfx, Rfb, PAS, Amx/Clv | Culture conversion at 4 weeks, persistent negative cultures taken bi-monthly until tx completion and follow-up of at least 2 years |
| **Kulkarni et al (2009) [38]** | Not specified for XDR TB regimen | Individualized based on DST results. Drugs used: Ofx, Km, Clr, PAS | Km | Ofx | R, H, S, E, Amk, Cs, Eto | Culture negative on gastric aspirates at 8 weeks, persistent negative cultures until tx completion, clinical and radiologic improvement (reduction of splenic abscess; resolution of brain tuberculomas) |
| **Schaaf et al (2009) [39]** | 19 mos of XDR TB treatment, including 18 mos after culture conversion | Individualized based on DST; Drugs used: Amx/Clv, Clr, Eto, Lzd, Z | None | None | Amk, H, Ofx, Cm, E, Clr, Trd | Culture negative at 1 month, persistent negative cultures taken bi-monthly until tx completion and follow-up, clinical and radiologic improvement (weight gain, reduced lung fibrosis) |
| **Schluger et al (1996) [40]** | At least 12 mos of pre-XDR TB treatment | Individualized based on DST results. Drugs used: Amk, Eto, Z | Amk | None | H, R, E, Cs, Cfx | Culture negative at 12 months, persistent negative cultures until tx completion, clinical and radiologic improvement (height/weight gain, regressed lymphadenopathy) |
| | | | | | H, R, Amk, S, Km | Tx completion, clinical and radiologic improvement (better overall health, reduced calcification in lungs, dissolution of lung infiltrate) |

Note

*Represents median or average length of treatment for population–based studies, and the total length of treatment for case–based studies. †Patient remains on treatment, including only case 2 from Shah et al (2011) [36]. DST: Drug sensitivity testing, CSF: Cerebrospinal fluid, Tx: Treatment, NS: Not specified, NA: not applicable, PA: Patient, FQ: fluoroquinolone, H: isoniazid, R: rifampicin, E: ethambutol, Z: Pyrazinamide, Rfb: rifabutin, Km: kanamycin, Amk: amikacin, Cm: capreomycin, S: streptomycin, Lfx: levofloxacin, Mfx: moxifloxacin, Ofx: ofloxacin, Cfx: Ciprofloxacin, Gfx: Gatifloxacin, Eto: ethionamide, Pto: protionamide, Cs: cycloserine, Trd: terizidone, PAS: p–aminosalicylic acid, Cfz: clofazimine, Lzd: linezolid, Amx/Clv: amoxicillin/clavulanate, Thz: thioacetazone, Clr: clarithromycin Clr, Ipm: imipenem, High–dose H: high dose isoniazid, Th: Thiacetazone, Rpt: Rifapentine, Pto: Prothionamide, Rif: Rifampin, Mem: Meropenem, M–C: Meropenem–Clavulanate, Ipm/Cln: imipenem/cilastatin.

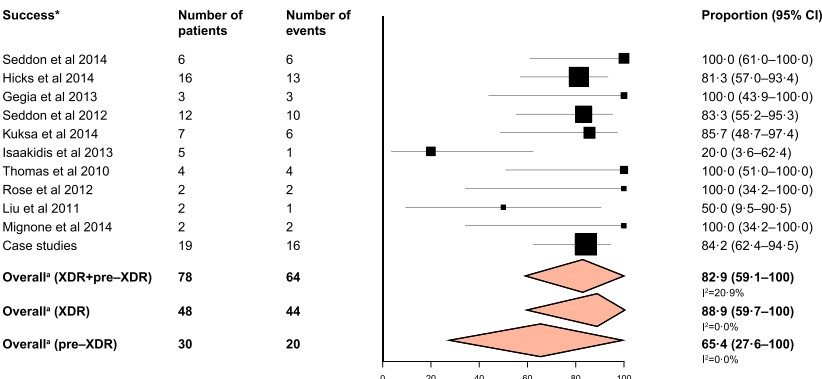

A: Proportion of XDR and pre–XDR TB patients with treatment success

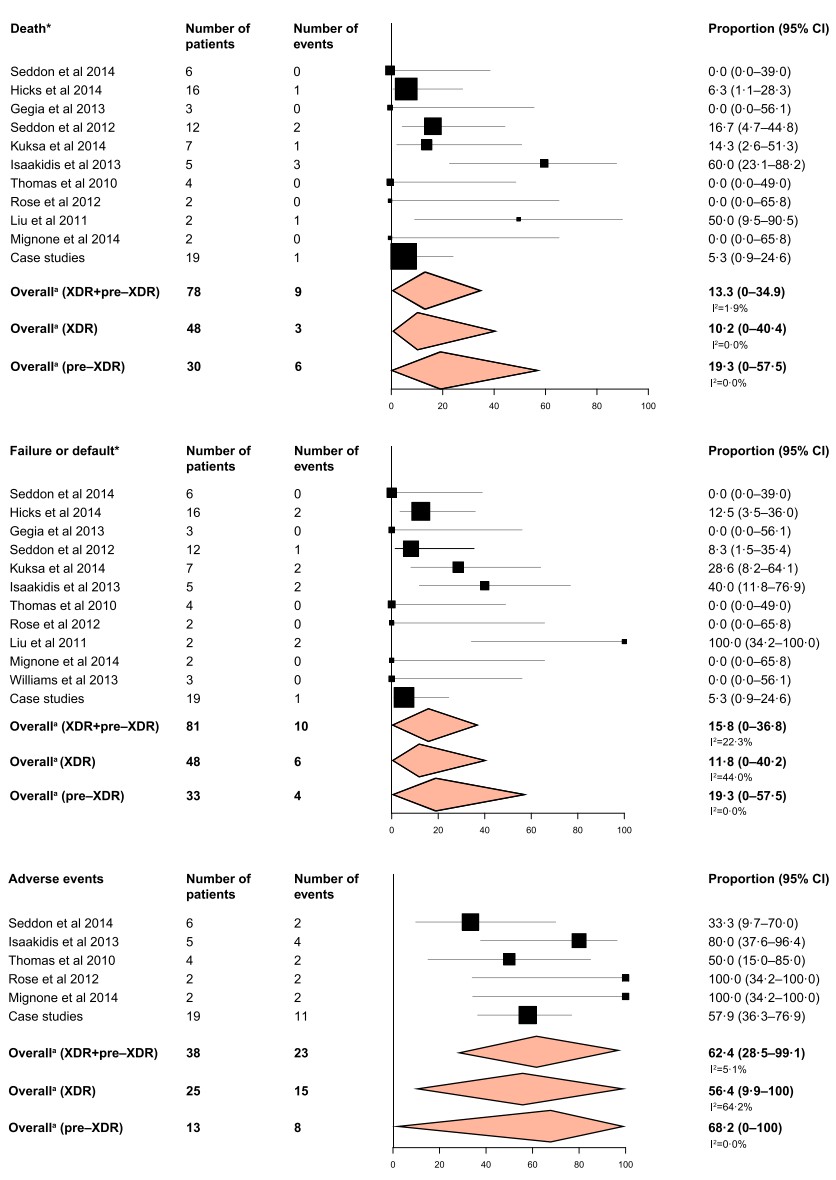

B: Proportion of XDR and pre–XDR TB patients experiencing death, failure or default, or adverse events

Note: *Excludes studies with unknown treatment outcomes. ª Bayesian random effects meta–analysis. Box sizes are proportional to the weights from the random effects model.

**Fig 2. Proportion of XDR and pre–XDR TB patients.** Note: (A) Patients with treatment success. (B) Patients experiencing death, failure or default, or adverse events. †Excludes studies with unknown treatment outcomes. [a] Random effects meta–analysis. [b] Bayesian random effects meta–analysis. ‡Personal communication with the principle investigator (Seddon, James. Conversation with: J Patra. 2015 December 08, 17).

comparison group for these variables; however, we found high treatment success rates (75–77%) for each of these subgroups. Meta–regression analyses also revealed similar results when analyses were stratified between XDR–and pre–XDR–TB cases (data not shown). We found low heterogeneity ($I^2 = 0$ to 50%), for most subgroups, except in patients having a contact with another MDR/XDR TB patient who showed moderate heterogeneity ($I^2 = 69 \cdot 7\%$).

## Adverse events

A large proportion of children with both XDR–and pre–XDR–TB experienced drug adverse events (XDR–TB, 56·4% 95%CI: 9·9–100; pre–XDR TB, 68·2% 95%CI: 0–100) (Figs 2B and S1). These included neurological effects (peripheral neuropathy, hearing loss) most commonly, followed by gastrointestinal (vomiting), endocrine hypothyroidism, hypokalemia), and haematological effects (anemia, neutropenia), with toxicity ranging from relatively mild (Grade 1) to life–threatening (Grade 5) (Table 3) [41]. The most commonly implicated drug was linezolid.

## Quality

Using a modified Newcastle–Ottawa scale to assess study quality, we found most population-based studies and case studies had a low and medium risk for selection bias, respectively (S2 Table). All included studies had a low risk of measurement bias, and most studies had a medium risk for bias in the assessment of outcomes.

## Discussion

Our pooled meta–analysis suggests moderate to high treatment success rates for pediatric patients with pre–XDR or XDR–TB. Our results for XDR–TB are comparable to outcomes for MDR–TB (81·7%; 95%CI: 72·5–90·8), while success estimates for pre–XDR–TB are lower [42]. Favorable treatment outcomes have been attributed to DST–guided treatment, the inclusion of injectable medications, and higher–generation fluoroquinolones (Lfx, Mfx and Gfx) [43]. All of these factors were near universal in the cases in our analysis. Equally important was contact tracing and initiating treatment early based on the source case's DST results. This has implications for case management; implementing these treatment guidelines may contribute to positive outcomes in children. Similar to adults with XDR–vs. MDR–TB, children in our sample experienced higher rates of death, failure, and loss to follow–up than their MDR–TB counterparts (5·9%; 95%CI: 1·3–10·5; 6·2%; 95%CI: 2·3–10·2, respectively) [42,44]. Rates of adverse events were also higher, more than half compared to a third of the pediatric cases with MDR–TB (39·1%; 95%CI: 28·7–49·4) [42]. HIV co–infection and severe adverse events was commonly reported in the patients who died [16,20,32]. Treatment for XDR–or pre–XDR–TB in addition to the antiretroviral therapy (ART) has been linked to severe adverse effects, perhaps owing to the higher generation fluoroquinolones, injectable drug use and linezolid, either alone or in combination with first–line anti–TB drugs or antiretroviral drugs [16,20,23,30–32,34–37,45]. Non–compliance and loss to follow-up is therefore a common occurrence, often leading to worsening of the condition, severe complications (such as brain tuberculoma, liver failure), subsequent treatment failure and death. This suggests a careful consideration of dosing regimens, duration of treatment, and monitoring strategies to optimize treatment efficacy,

**Table 3. Adverse events.**

| Study | Adverse events (n, %)* | Grade | Implicated drugs or health conditions |
|---|---|---|---|
| **Population–based** | | | |
| Hicks et al (2014) [16] | Ototoxity<br>Hypersensitivity | Unclear | Amk/Km/Cm<br>Rif |
| Kuksa et al (2014) [17] | NS | NA | NA |
| Mignone et al (2014) [18] | Vomiting, leg pain (1, 50%)<br>Myelotoxicity (1, 50%) | Unclear | Eto<br>NS |
| Seddon et al (2014)† [24] | Lethargy (1, 17%) | 1 | |
| | Multiple side effects (appetite, vomiting, itch, rash) (1, 17%) | 3,1,1,1 | NS |
| Isaakidis et al (2013) [20] | Convulsions (1, 20%)<br>GI Intolerance (2, 40%)<br>Peripheral neuropathy (3, 67%)<br>Hearing loss (1, 20%)<br>Hypokalemia (1, 20%)<br>Hypothyroidism (1, 20%) | Unclear | Cs<br>Eto, Amk D4T, Cs, Eto Km<br>Cm Eto |
| Gegia et al (2013) [21] | Not evaluated | NA | NA |
| Williams et al (2013) [22] | NS | NA | NA |
| Rose et al (2012) [23] | Pancreatitis (2, 40%), Lactic acidosis (1, 20%) | Unclear | Lzd |
| Seddon et al (2012) [24] | Not evaluated | NA | NA |
| Liu et al (2011) [25] | NS | NA | NA |
| Thomas et al (2010) [26] | Acute psychosis (1, 25%)<br>Hyperactivity and mood lability (1, 25%) | Unclear | Cs, EFV<br>Cs |
| **Case-studies** | | | |
| Salazar–Austin et al (2015) [27] | Hypothyroidism (1, 100%)<br>Bronze skin colouration (1, 100%) | Unclear | PAS<br>Cfz |
| Alsleben et al (2014) [28] | Hepatitis (1, 100%)<br>Hypothyroidism, mild anemia and gastrointestinal symptoms (1, 100%) | Unclear<br>Unclear 1 | PAS, Eto, Z NS |
| Mohan et al (2014) [29] | NS | | NA |
| Rodrigues et al (2014) [30] | Progressive anemia, Medullary Hypoplasia Edema of knees, thigh paresthesias, arthralgia, synovitis (1, 100%) | Unclear 3 | Lzd Cfx |
| Uppuluri et al (2014) [31] | Mild to moderate hearing loss (1, 100%) | 2 | Amk |
| Katragkou et al (2013) [32] | Mild neutropenia (1, 50%) | 1 | Lzd, Rfb |
| | Acute hepatic insufficiency and subsequent death (1, 50%) | 5 | Ct, 3[rd] generation cephalosporins, Rif |
| Payen et al (2012) [33] | None | NA | NA |
| Dauby et al (2011) [34] | Moderate peripheral neuropathy (1, 100%) | 2 | Lzd |
| Kjollerstrom et al (2011) [35] | Peripheral neuropathy (2, 100%) | 3 | Lzd |
| | Anemia (1, 50%) | 3 | Lzd, Sickle cell disease |
| Shah et al (2011) [36] | Moderate to severe mixed hearing loss (1, 33%) | 2 to 3 | Amk |

(*Continued*)

**Table 3.** (Continued)

| Study | Adverse events (n, %)* | Grade | Implicated drugs or health conditions |
|---|---|---|---|
| Anger et al (2010) [37] | Haematological (13, 81%)<br>GI (13, 81%)<br>Neurological (7, 44%) | Unclear | Lzd |
| Kulkarni et al (2009) [38] | NS | NA | NA |
| Schaaf et al (2009) [39] | None | NA | NA |
| Schluger et al (1996) [40] | NS | NA | NA |

Note

*Number and proportion of XDR cases. †Personal communication with the clinical investigator (Seddon, James. Conversation with: Jay Patra. 2015 December 08, 17.). Tx: Treatment, DST: Drug sensitivity testing, CSF: Cerebrospinal fluid, N A: Not applicable, NS: Not specified, GI: Gastrointestinal, EFV: efavirenz, D4T: stavudine, Ct: Colistin, Z: Pyrazinamide, Rfb: rifabutin, Km: kanamycin, Amk: amikacin, Cm: capreomycin, Cfx: Ciprofloxacin, Gfx: Gatifloxacin, Eto: ethionamide, Pto: protionamide, Cs: Cycloserine, Trd: terizidone, PAS: p-aminosalicylic acid, Cfz: clofazimine, Lzd: linezolid.

minimize complications, and encourage treatment compliance. To optimize outcomes, it is therefore essential to implement and emphasize the importance of directly observed therapy to enhance adherence and mitigate the risk of treatment failure, particularly given the observed higher rates of non- compliance and loss to follow-up.

An additional concern is the social stigma in reporting HIV or TB that can delay or prevent the initiation of treatment [20]. Social acceptance and support for affected individuals is therefore just as important in preventing poor outcomes, a goal that can be achieved through social counseling and education to prevent delays in treatment initiation and improve patient acceptance. Treatment success was also high in patients treated with injectables and fluoroquinolones; however all patients in these subgroups completed therapy, and upon follow–up, none showed relapse or permanent adverse effects. This further speaks to the importance of direct observation and treatment compliance with subsequent follow–up to ensure continual care in case of relapse or adverse events. Children in our study showed higher success rates than adult XDR–TB cases, described in a previous meta–analysis (43·7%; 95%CI: 32·8–54·5) [46]. Similar trends have been found in previous cohorts, with children showing 20–50% more success, and much lower mortality, failure or default on treatment than adults, although the differences were not significant, perhaps because of the significantly smaller sample of pediatric cases [17,25].

## Strengths and limitations

Our study has several strengths. To our knowledge, this is the first meta–analysis exploring treatment outcomes for pre–XDR–and XDR–TB in children. We also found low heterogeneity in our analysis, which highlights the consistency of treatment outcomes across studies and sites. In addition to cohort studies, we also report on outcomes for individual case studies, which provide insight into the specific aspects of treatment that may contribute to success or unfavorable outcomes. Almost all cases in our analysis were culture–confirmed, which suggests a very low possibility for misdiagnosis. Though culture–confirmation is considered the most reliable method of diagnosis, the three cases who were diagnosed based on their contact

history perhaps represent a wider spectrum of pediatric cases, who (given the paucibacillary nature of much of pediatric TB) are more commonly diagnosed based on household contact with an MDR–TB patient [47]. All culture–confirmed children were also treated based on their DST results, with treatment regimens strengthened as recommended by guidelines for better outcomes [3]. Additionally, in this review, while certain studies were excluded from the meta- analysis due to methodological limitations, they were still included, separately, (S1–S3 Tables) [19,24,48–68] to provide a comprehensive overview of existing literature.

Our study also has several limitations. Since our analysis primarily consisted of cohort studies, and a number of case-series with a relatively small sample size, inherently reduces the statistical power necessary to generalize our findings to the broader pediatric population with pre–XDR–and XDR–TB. However, considering the challenges of diagnosing pediatric XDR–TB, the high-cost associated with treatment complexity, limited availability of second-line pediatric-approved drugs, and the prolific incidence of adverse events reported in adults, a lack of large–scale studies in this population is anticipated [1,47]. To address this methodological challenge, a Bayesian approach was employed, which is particularly well-suited for analyzing data with smaller sample sizes, allowing for the preservation of statistical power and precision in parameter estimates.

The inclusion of both prospective and retrospective studies poses another key limitation as both study designs are susceptible to their own unique biases. On one hand, retrospective studies rely primarily on historical self-reported data by participants, physicians, and/or researchers, and are prone to selection and recall bias; on the other hand, prospective studies, while generally more robust, may still be subject to temporal bias owing to their time-dependent confounding variables. There may also have been under-reporting of adverse events by patients and their families, thereby introducing reporting bias in the study results, potentially leading to an overestimation of positive outcomes. Furthermore, the revision of the defining diagnostic criteria for pre-XDR TB and XDR TB by the WHO in 2021 may also have led to inconsistencies in findings reported across studies, with this classification bias consequently affecting the reliability of the reported effect sizes and the low, yet noticeable statistical heterogeneity of the pooled data [69].

A sensitivity analysis was also performed to assess the robustness and reliability of our study's findings (S6 Table). However, due to the limited number of data points for certain outcomes, we were unable to conduct meta-regression analyses for outcomes other than treatment success. Consequently, our sensitivity (subgroup) analyses were primarily restricted to examining the treatment success rate alone to maintain the robustness of our findings.

Additionally, to assess the impact of publication bias in the included studies, we incorporated funnel plots based on study success rates using proportion data (S1A Fig, S7 Table). For the outcomes of failure and death, where initial data on denominators were not available, we estimated sample sizes using the confidence intervals of the success proportions and generated funnel plots for these outcomes (S1B and S1C Fig, S7 Table). Due to insufficient data, funnel plots for adverse events could not be generated. However, funnel plots for death rates and failure/default rates have been included. These additional analyses enhance our assessment of potential publication bias and strengthen the reliability of our meta-analysis findings.

Another limitation of our study is the presence of missing data for key outcomes.

Specifically, missing data were limited for treatment success (5 studies), death (5 studies), and treatment failure (3 studies). However, there was a more substantial amount of missing data for adverse events reported in the included studies, which may impact the completeness and reliability of the findings. To address this, a detailed description of the reported adverse events and a measure of their relative severity, defined with a grade score between 1 to 4, is provided in S5 Table. Additionally, the use of Bayesian inference in our analysis was beneficial

in this regard because it can handle missing data naturally, without requiring imputation or deletion.

This is because the missing data are treated as unobserved parameters that can be estimated from the data and prior settings.

Furthermore, despite the increasing uptake of novel therapies such as delamanid and bedaquiline for the treatment of MDR-TB and XDR-TB, offering favourable outcomes in adults, there remains limited evidence at present to support its use in pediatric patients; hence, why our study did not consider the use of these recently approved drugs in our analysis [70].

Nonetheless, we summarize here the best available evidence, to date, on the treatment outcomes for pre–XDR–and XDR–TB in children and adolescents. Our subgroup analysis did not show any significant results, and, as such, we were unable to draw any conclusive inferences regarding the impact of several key clinical and sociodemographic features that have been found significant for treatment outcomes in adults. These include HIV prevalence, the patient's previous treatment, contact history, duration of treatment, use of injectables and use of fluoroquinolones [46]. We did not explore whether the outcomes for children with pre–XDR and XDR–TB were significantly different from those with MDR–TB, or from adults with XDR–TB, and consider it worthwhile to pursue this analysis using larger cohorts from survey–based data or clinical trials.

## Conclusion

In conclusion, our study and analysis highlight several important considerations for the management of pediatric patients with pre-XDR and XDR-TB. Here, we highlight certain unique aspects of successful treatment regimen, as well as the drugs implicated in the development of adverse events. Given the limited generalizability of our analysis, there is a need for further analysis using nationally or sub–nationally representative data. Additional exploration of the impact of HIV status, previous treatment and contact history, and the use of specific drugs on the treatment outcomes will be helpful in further strengthening and improvising the treatment guidelines for pre–XDR–and XDR–TB in the pediatric population. Overall, the targeted strategies discussed in this review and meta-analysis, ranging from individualized treatment plans to comprehensive follow-up care, are vital for improving health outcomes and advancing the management of pre-XDR and XDR-TB in children and adolescents.

## Supporting information

**S1 Table. Additional clinical characteristics of XDR TB patients.** Note: †Data represents all XDR cases of all ages as the information was not specified for pediatric cases only. PTB: Pulmonary tuberculosis, EPTB: Extrapulmonary tuberculosis, NS: Not specified.
(PDF)

**S2 Table. Quality assessment using Newcastle–Ottawa scale.** Note: (A) Quality assessment of studies included in the meta-analysis. (B) Quality assessment of studies excluded in the meta-analysis due to missing data on key outcomes such as adverse events and treatment regimens, and data points required to calculate pooled proportions. These studies were included to provide a comprehensive review of current literature. ‡ Long term follow-up is classified as any treatment duration > 12 months; NS: Not specified; NA: Not available.
(PDF)

**S3 Table. Description of included studies and demographic characteristics of patients Note: The data above is for studies excluded from the meta-analysis, but included to ensure extensive survey of current literature.** *Represents the overall sample, including

MDR/DRTB cases; information for XDR cases is shown if specified in the study. †The majority of XDR cases were culture–positive and were treated based on their DST results except for the cases in Seddon et al (2014), where 50% of the XDR cases were culture–confirmed, and thus, DST was completed on 50% of cases' samples; the remaining 50% cases were treated based on the DST results from the source. ‡HIV status was not known (Williams et al (2013)) or tested (Gegia et al (2013)) for all XDR cases. XDR: Extensively drug resistant, P cohort: Prospective cohort, R cohort: Retrospective cohort, NS: Not specified, DST: Drug susceptibility test, TST: Tuberculosis Skin Test, CXR: Chest X–Ray, CT scan: Computerized Tomography.
(PDF)

**S4 Table. Description of treatment regimen.** Note: The data above is for studies excluded from the meta-analysis, but included to ensure extensive survey of current literature. *Represents median or average length of treatment for population–based studies, and the total length of treatment for case–based studies. †Patient remains on treatment, including only case 2 from Shah et al (2011). DST: Drug susceptibility testing, CSF: Cerebrospinal fluid, Tx: Treatment, NS: Not specified, NA: Not applicable, FQ: Fluoroquinolone, H: Isoniazid, R: Rifampicin, E: Ethambutol, Z: Pyrazinamide, Rfb: Rifabutin, Km: Kanamycin, Amk: Amikacin, Cm: Capreomycin, S: Streptomycin, Lfx: Levofloxacin, Mfx: Moxifloxacin, Ofx: Ofloxacin, Cfx: Ciprofloxacin, Gfx: Gatifloxacin, Eto: Ethionamide, Pto: Protionamide, Cs: Cycloserine, Trd: Terizidone, PAS: p–aminosalicylic acid, Cfz: Clofazimine, Lzd: Linezolid, Amx/Clv: Amoxicillin/clavulanate, Thz: Thioacetazone, Clr: Clarithromycin Clr, Ipm: Imipenem, High–dose H: High dose isoniazid, Th: Thiacetazone, Rpt: Rifapentine, Pto: Prothionamide, Rif: Rifampin, Mem: Meropenem, M–C: Meropenem–Clavulanate, Ipm/Cln: Imipenem/cilastatin.
(PDF)

**S5 Table. Adverse events.** Note: The data above is for studies excluded from the meta-analysis, but included to ensure extensive survey of current literature. *Number and proportion of XDR cases. †Personal communication with the clinical investigator (Seddon, James. Conversation with: Jay Patra. 2015 December 08, 17.). Tx: Treatment, DST: Drug susceptibility testing, CSF: Cerebrospinal fluid, N A: Not applicable, NS: Not specified, GI: Gastrointestinal, EFV: Efavirenz, D4T: Stavudine, Ct: Colistin, Z: Pyrazinamide, Rfb: Rifabutin, Km: Kanamycin, Amk: Amikacin, Cm: Capreomycin, Cfx: Ciprofloxacin, Gfx: Gatifloxacin, Eto: Ethionamide, Pto: Protionamide, Cs: Cycloserine, Trd: Terizidone, PAS: p–aminosalicylic acid, Cfz: Clofazimine, Lzd: Linezolid.
(PDF)

**S6 Table. Pooled treatment success among various subgroups of patients (sensitivity analysis).** Note: FQ: Fluroquinolone, DST: Drug susceptibility testing, MDR: Multi–drug resistant, XDR: Extensively drug resistant, ref: Reference category.
(PDF)

**S7 Table. Treatment percent (%) success, failure/default, and death.**
(PDF)

**S1 Fig. Funnel plot assessing publication bias in studies reporting (A) treatment success rates, (B) failure/default rates, and (C) death rates.**
(EPS)

**S1 Text. PRISMA checklist for the meta–analysis.**
(PDF)

**S2 Text. Study protocol for systematic review and meta–analysis to identify treatment outcomes among children and adolescents with extensively drug–resistant (XDR) and pre–XDR tuberculosis.**
(PDF)

**S1 Appendix. All study and patient-related characteristics were extracted using the data extraction tool developed in excel.**
(XLSX)

## Acknowledgments

We thank Dr. James Seddon (Imperial College, UK) and Dr. Simon Schaaf (Stellenbosch University, South Africa) for their valuable comments and suggestions on previous draft of this manuscript.

## Author Contributions

**Conceptualization:** Jayadeep Patra, Jurgen Rehm.

**Data curation:** Jayadeep Patra, Pranshu Maini, Jady Liang, Anwesh Patra.

**Formal analysis:** Jayadeep Patra, Hyacinth Irving.

**Investigation:** Jayadeep Patra, Pranshu Maini, Jady Liang.

**Methodology:** Jayadeep Patra.

**Project administration:** Jayadeep Patra.

**Resources:** Jayadeep Patra.

**Software:** Jayadeep Patra, Hyacinth Irving.

**Supervision:** Jayadeep Patra, Mandar Paradkar, Jurgen Rehm.

**Validation:** Jayadeep Patra.

**Visualization:** Jayadeep Patra, Pranshu Maini.

**Writing – original draft:** Jayadeep Patra.

**Writing – review & editing:** Jayadeep Patra, Pranshu Maini, Mandar Paradkar.

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
