## [Decision Letter · Decision Letter 0]

26 Jun 2024

PGPH-D-24-01097

Treatment outcomes among children and adolescents with extensively drug–resistant (XDR) and pre–XDR tuberculosis: systematic review and meta–analysis

Dear Dr. Patra,

Thank you for submitting your manuscript to PLOS Global Public Health. After careful consideration, we feel that it has merit but does not fully meet PLOS Global Public Health’s publication criteria as it currently stands. Therefore, we invite you to submit a revised version of the manuscript that addresses the points raised during the review process.

This systematic review and meta-analysis on paediatric pre-XDR and XDR TB treatment outcomes is a commendable attempt and an important topic. However, some major issues such as the variability in the management of pre-XDR and XDR-TB with inclusion of newer drugs (Bedaquiline, Delaminid) and newer regimens (BPaL) across the studies and time need to be addressed. Please also mention how the changing definitions would have an impact on the treatment outcomes studied. Addressing the comments of the reviewers on the methodological aspects will also add value.

We look forward to receiving your revised manuscript.

Kind regards,

Sonali Sarkar

Academic Editor

Journal Requirements:

1. Please provide separate figure files in .tif or .eps format only and remove any figures embedded in your manuscript file. Please also ensure all files are under our size limit of 10MB.

Reviewers' comments:

Reviewer's Responses to Questions

**Comments to the Author**

1. Does this manuscript meet PLOS Global Public Health’s publication criteria? Is the manuscript technically sound, and do the data support the conclusions? The manuscript must describe methodologically and ethically rigorous research with conclusions that are appropriately drawn based on the data presented.

Reviewer #1: Yes

Reviewer #2: Yes

Reviewer #3: Yes

2. Has the statistical analysis been performed appropriately and rigorously?

Reviewer #1: Yes

Reviewer #2: Yes

Reviewer #3: Yes

3. Have the authors made all data underlying the findings in their manuscript fully available (please refer to the Data Availability Statement at the start of the manuscript PDF file)?

Reviewer #1: Yes

Reviewer #2: Yes

Reviewer #3: Yes

4. Is the manuscript presented in an intelligible fashion and written in standard English?

Reviewer #1: Yes

Reviewer #2: Yes

Reviewer #3: Yes

5. Review Comments to the Author

Reviewer #1: The systematic review estimated the pooled frequency of treatment outcomes (such as cure, treatment completion, death, and adverse events) among children with pre-XDR or XDR TB. The topic is very interesting and holds significant clinical importance. However, I have several major comments:

a. Publication Dates of Included Studies: All included studies were published before 2015. This means that novel drugs such as bedaquiline and delamanid were not used in these studies, and some new treatment options (such as BPaL) were also not considered. This inherent limitation may lead to an underestimation of the conclusions.

b. Sample Size and Treatment Complexity: The sample sizes of each included study are too small, which could affect the accuracy of the findings. Additionally, the complexity of treatment choices should not be ignored, as it significantly impacts the assessment of treatment outcomes.

c. Revised Definitions of Pre-XDR and XDR TB: The definitions of pre-XDR and XDR TB were revised in 2021. This limits the value of the conclusions, as they cannot be generalized to the current situation.

Reviewer #2: 1. The introduction provides a good overview of the prevalence and significance of XDR and pre-XDR tuberculosis. However, it could be strengthened by adding more detail on the specific challenges of diagnosing and treating these conditions in pediatric populations.

2. Why no mention about publication bias based on funnel plot or doi plot. Kindly do grade analysis to give a more reliable met analysis.

3. Sensitivity analysis component is also missing as a part of met analysis.

4. The reporting of adverse events is crucial. It would be helpful to categorize these adverse events by severity to better understand the treatment burden.

Reviewer #3: Introduction:

1. The introduction could be expanded to provide further insights into the burden of TB among children and adolescents. This section could detail the proportion of TB cases in children that are pulmonary versus extrapulmonary, as well as the breakdown of drug-sensitive TB cases. Additionally, it could include statistics on drug-resistant TB, specifying the incidence of multidrug-resistant (MDR), pre-extensively drug-resistant (pre-XDR), and extensively drug-resistant (XDR) TB

2. The section on treatment benefits from a more comprehensive overview of the various treatment options currently practiced worldwide. This could include standard treatment protocols, variations in treatment strategies across different regions, and the latest advancements in TB treatment.

Methodology:

1. List the specific inclusion and exclusion criteria used for selecting studies like extrapulmonary TB cases included or excluded.

2. Details on the data extraction process – any specific template used may be mentioned for reproducibility

3. Describe how missing data were handled in the analysis, particularly for key outcomes.

4. The R^ value may be reported for Brooks–Gelman criteria to understand the model convergence in Monte Carlo Markov chain simulations models.

Results:

1. Results of sub-group analysis particularly on different treatment regiments and its effect on treatment outcomes can add more value for the study and address the interest of the readers

Discussion:

1. Discuss the findings in a broader context like how they align with or challenge current TB treatment guidelines and policies for children and adolescents

Limitations:

1. Elaborate on specific limitations related to study designs, such as retrospective versus prospective studies, and the impact of varying definitions and diagnostic criteria across studies.

2. Discuss the potential impact of publication bias and describe any statistical tests (e.g., Egger's test) used to assess it.

Recommendations:

1. Provide recommendations for clinical practice based on the study findings, such as potential adjustments to treatment regimens or monitoring protocols for this age group.

6. PLOS authors have the option to publish the peer review history of their article (what does this mean?). If published, this will include your full peer review and any attached files.

**Do you want your identity to be public for this peer review?** For information about this choice, including consent withdrawal, please see our Privacy Policy.

Reviewer #1: No

Reviewer #2: No

Reviewer #3: No

---

## [Decision Letter · Decision Letter 1]

10 Oct 2024

Treatment outcomes among children and adolescents with extensively drug–resistant (XDR) and pre–XDR tuberculosis: systematic review and meta–analysis

PGPH-D-24-01097R1

Dear Dr. Patra,

We are pleased to inform you that your manuscript 'Treatment outcomes among children and adolescents with extensively drug–resistant (XDR) and pre–XDR tuberculosis: systematic review and meta–analysis' has been provisionally accepted for publication in PLOS Global Public Health.

Best regards,

Sonali Sarkar

Academic Editor

Reviewer Comments (if any, and for reference):

Reviewer's Responses to Questions

**Comments to the Author**

1. If the authors have adequately addressed your comments raised in a previous round of review and you feel that this manuscript is now acceptable for publication, you may indicate that here to bypass the “Comments to the Author” section, enter your conflict of interest statement in the “Confidential to Editor” section, and submit your "Accept" recommendation.

Reviewer #1: All comments have been addressed

Reviewer #2: All comments have been addressed

Reviewer #3: All comments have been addressed

2. Does this manuscript meet PLOS Global Public Health’s publication criteria? Is the manuscript technically sound, and do the data support the conclusions? The manuscript must describe methodologically and ethically rigorous research with conclusions that are appropriately drawn based on the data presented.

Reviewer #1: Yes

Reviewer #2: Partly

Reviewer #3: Yes

3. Has the statistical analysis been performed appropriately and rigorously?

Reviewer #1: Yes

Reviewer #2: Yes

Reviewer #3: Yes

4. Have the authors made all data underlying the findings in their manuscript fully available (please refer to the Data Availability Statement at the start of the manuscript PDF file)?

Reviewer #1: Yes

Reviewer #2: Yes

Reviewer #3: Yes

5. Is the manuscript presented in an intelligible fashion and written in standard English?

Reviewer #1: Yes

Reviewer #2: Yes

Reviewer #3: Yes

6. Review Comments to the Author

Reviewer #1: (No Response)

Reviewer #2: Though most of the comments have been addressed, the funnel plots are not showing any symmetry and should be verified by eggers or beggs test and interpretation of the results to be given in detail.

Reviewer #3: (No Response)

7. PLOS authors have the option to publish the peer review history of their article (what does this mean?). If published, this will include your full peer review and any attached files.

**Do you want your identity to be public for this peer review?** For information about this choice, including consent withdrawal, please see our Privacy Policy.

Reviewer #1: **Yes: **Mao-Shui Wang

Reviewer #2: No

Reviewer #3: No
